# Perifascial areolar tissue graft promotes angiogenesis and wound healing in an exposed ischemic component rabbit model

**Toru Miyanaga**[1]*, **Yasuo Yoshitomi**[2], **Aiko Miyanaga**[3]

1 Department of Plastic Surgery, Kanazawa Medical University, Kahoku, Ishikawa, Japan, 2 Department of Biochemistry, Kanazawa Medical University, Kahoku, Ishikawa, Japan, 3 Department of Nursing, Kanazawa Medical University, Kahoku, Ishikawa, Japan

* miyanaga@kanazawa-med.ac.jp

**Data Availability Statement:** All relevant data are within the manuscript and its Supporting Information files.

## Abstract

Multiple studies have reported the use of perifascial areolar tissue (PAT) grafts to treat wounds involving exposed ischemic tissues, avascular structures, and defective membrane structures. Our objective was to assess the quantitative effects of PAT grafts and their suitability for wounds with ischemic tissue exposure and to qualitatively determine the factors through which PAT promotes wound healing and repair. We conducted histological, immunohistochemical, and mass spectrometric analyses of the PAT grafts. PAT grafts contain numerous CD34+ progenitor/stem cells, extracellular matrix, growth factors, and cytokines that promote wound healing and angiogenesis. Furthermore, we established a male rabbit model to compare the efficacy of PAT grafting with that of an occlusive dressing treatment (control) for wounds with cartilage exposure. PAT grafts could cover ischemic components with granulation tissue and promote angiogenesis. Macroscopic and histological observations of the PAT graft on postoperative day seven revealed capillaries bridging the ischemic tissue (vascular bridging). Additionally, the PAT graft suppressed wound contraction and alpha smooth muscle actin (αSMA) levels and promoted epithelialization. These findings suggested that PAT can serve as a platform to enhance wound healing and promote angiogenesis. This is the first study to quantify the therapeutic efficacy of PAT grafts, revealing their high value for the treatment of wounds involving exposed ischemic structures. The effectiveness of PAT grafts can be attributed to two primary factors: vascular bridging and the provision of three essential elements (progenitor/stem cells, extracellular matrix molecules, and growth factors/cytokines). Moreover, PAT grafts may be used as transplant materials to mitigate excessive wound contraction and the development of hypertrophic scarring.

## 1. Introduction

Wounds with exposed ischemic or avascular tissues, such as tendons, bone, cartilage, and artificial implants or plates, often cannot be covered entirely with granulation tissue upon using

**Funding:** This research received support from the Japan Society for the Promotion of Science (JSPS) KAKENHI Grant Number 19K12809 and an SBC Research Support Grant. The funders had no role in study design, data collection and analysis, decision to publish, or preparation of the manuscript.

**Competing interests:** The authors have declared that no competing interests exist.

conservative treatment as they require surgical treatment such as skin flaps. In cases where surgical intervention is more invasive, it is preferable to allow for granulation on ischemic and avascular structures, especially in older individuals. This can help achieve wound closure through conservative treatment or a less invasive skin grafting technique. Several clinical studies have reported the potential use of perifascial areolar tissue (PAT) grafts in the treatment of such wounds [1–7]. Additionally, PAT grafts have demonstrated versatility as reconstructive membranes in tympanoplasty [8, 9] as well as for skull base or spinal dural reconstruction in cases of cerebrospinal fluid leakage [8–10]. Thus, PAT may serve as a high-quality graft tissue for wounds involving the exposed ischemic tissue and any membrane structure. Despite the success of PAT grafts in clinical case studies, fundamental research on their molecular composition and ability to promote wound healing in ischemic structures is lacking [3, 5, 6].

The primary objective of this study was to examine the effect of PAT grafts on wounds with exposed ischemic tissue, and the secondary objective was to investigate the molecular composition of PAT. Using histological and immunohistochemical analyses, we established a rabbit wound model with exposed cartilage to compare the wound healing process between PAT-grafted and untreated wounds and to determine the effectiveness of PAT grafts. Furthermore, PAT was comprehensively analyzed through histological and immunohistochemical studies, as well as through data-independent acquisition (DIA) mass spectrometry analysis, enabling the examination of the extracellular matrix, cellular composition, cytokines, and growth factors that influence the therapeutic effects of PAT grafts. Elucidating the effect of PAT on wounds with exposed ischemic tissue and understanding its survival and composition (including the extracellular matrix, cells, and growth factors) could facilitate the clinical application of PAT in patients with such wounds and contribute to membrane defect reconstruction.

## 2. Materials and methods

An overview of the experimental design is presented in Fig 1.

### 2.1 Animal models

The experimental protocols involving animals and their tissues were approved by the Ethics Committee of Kanazawa Medical University School of Medicine in accordance with the Animal Research Reporting of In Vivo Experiments (ARRIVE) guidelines (approval numbers: 2014–79, 2015–69, and 2020–65). This study used rabbits as a model organism because rabbit PAT has been identified in previous basic animal studies [11]. Male Japanese white rabbits (S1c:JW/CSK) were obtained from Sankyo Labo Service Corporation (Tokyo, Japan). A total of 15 rabbits, aged 14–16 weeks and weighing 2.5–3.5 kg, were individually housed in cages under a 12-hour light/dark cycle with ad libitum access to food and water. A 1-week acclimation period was used before commencing the experiments.

For anesthesia, the rabbits were administered pentobarbital (25 mg/kg; Somnopentyl®; Kyoritsu Seiyaku Corp., Tokyo, Japan) via the ear marginal vein. Subsequently, 0.5% lidocaine-containing epinephrine (Xylocaine; Sandoz K, Tokyo, Japan) was administered locally to the ears and lateral abdominal walls to reduce pain. The ears and lateral hypogastrium were shaved and a povidone–iodine solution was used for disinfection before the surgical procedure. The connective tissue with a vascular plexus, located just above the abdominal external oblique muscle, was identified as the PAT after making a channel-shaped incision (10 × 5 cm) in the hypogastric skin and elevating the skin flap. PAT samples (6 × 3 cm), including thin connective tissues, were collected (Fig 1A). A portion of harvested PAT from each rabbit was used as a wound autograft. After wound autografting, the remaining PAT from each rabbit was used for histological analysis and DIA mass spectrometry. Full-thickness skin defects

A

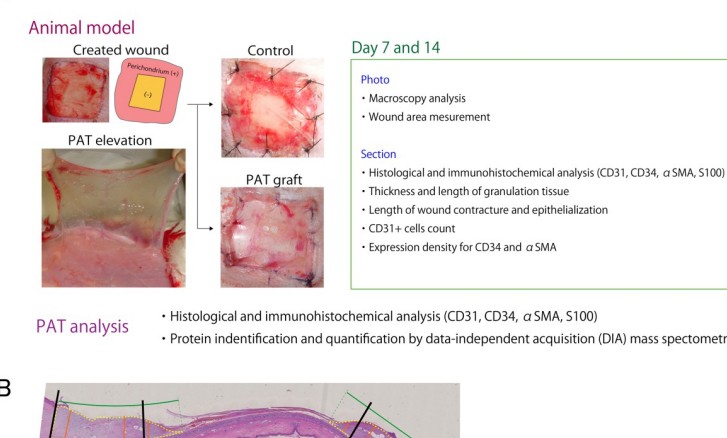

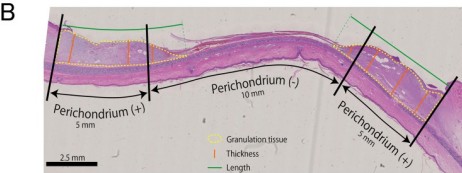

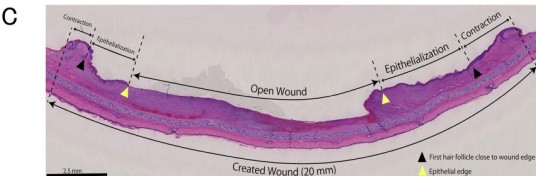

**Fig 1. Materials and methods. A.** A schematic representation of the materials and methods employed. PAT: perifascial areolar tissue. **B.** Measurement of granulation tissue. The central 10-mm area represents a perichondrium defect; the outer 10-mm area represents a healthy perichondrium. Yellow dotted line: granulation tissue; orange line: granulation tissue thickness; green line: length of granulation tissue. **C.** Measuring the open wound, wound contracture, and epithelialization. Black triangle: first hair follicle near the wound edge; yellow triangle: epithelial tip.

($2 \times 2$ cm) were created in the region near each ear, approximately 4 cm from the external meatus. The perichondrium was removed to expose the cartilage ($1 \times 1$ cm) at the center of the defect. Two wounds per rabbit were created as the operation was performed on both ears, and the wounds were randomly assigned to either the PAT or control group by independent assistants. Fifteen rabbits were used in the experiment. In the PAT group, an autograft was placed over the entire skin defect and sutured using 6–0 nylon. The graft was covered with a polyurethane film dressing (TegaDerm®; 3M, Minneapolis, MN, USA) and secured with 5–0 nylon. In the control group, grafting was not performed, and the wound was covered with a polyurethane film dressing (TegaDerm®; 3M) and secured with 5–0 nylon. The experimental duration per rabbit was approximately 3 h. The film dressing was removed from each wound at the end of the experimental period (7 or 14 days).

Subsequently, photographs of the wounds were captured using a Sony 20.1-megapixel camera (Sony, Tokyo, Japan) for macroscopic analysis and measurement of the open wound area. The rabbits were euthanized by administering an overdose of anesthetic (25 mg/kg pentobarbital; Somnopentyl®; Kyoritsu Seiyaku Corp., Tokyo, Japan) via the ear marginal vein. No humane endpoints were observed. The wound regions were excised, fixed in 10% buffered formalin, and embedded in paraffin. The exclusion criteria included instances in which animals died during the experimental phase or when the planned cartilage-exposed wound model was not achievable. This was particularly relevant in cases in which necrosis developed within the cartilage of the wound. Six wounds were analyzed per group at each designated endpoint.

Despite initially creating 30 wounds, one rabbit with two wounds did not survive the experimental period. Furthermore, four wounds exhibited sustained ear cartilage damage during surgery, resulting in necrosis.

Consequently, these wounds were excluded from the analysis. A total of 24 wounds were examined; 12 wounds (six each for control and PAT) were obtained 7 days after the intervention, and 12 wounds (six each for control and PAT) were obtained 14 days after the intervention. An independent assistant was responsible for blinding the results.

## 2.2 Histological and immunohistochemical analyses

In accordance with established protocols, paraffin-embedded sections were stained with hematoxylin and eosin (H&E) and Elastica van Gieson stain. Immunohistochemistry was performed through deparaffinizing and rehydrating the tissue sections. Subsequently, the xylene was removed using 100% ethanol. After that, 4-μm tissue sections were gradually hydrated using a graded alcohol series. The methods used for specific antigen retrieval and probing with primary monoclonal antibodies (mouse anti-cluster of differentiation 31 [CD31], rat anti-cluster of differentiation 34 [CD34], mouse anti-S100 protein, and mouse anti-α-SMA antibody) are described in Supplemental Digital Content 1. Endogenous peroxidase activity was neutralized using 3% hydrogen peroxide prepared in methanol, and 3,3-diaminobenzidine was used as the peroxidase substrate. Mayer's hematoxylin staining was performed to visualize the cell nuclei in the specimens, as per the standard laboratory protocol.

## 2.3 Open wound area measurement

At the end of the experiment, digital photographs were taken to capture the open wound areas in each group. The open wound area was measured using ImageJ software ver. 2.0.1 (SPSS National Institutes of Health, Bethesda, MD, USA) according to a previously published procedure [12].

## 2.4 Thickness and length of the granulation tissue

To examine the general healing process in wounds with exposed cartilage, we compared the granulation ability of wounds with and without the perichondrium in the control group. Subsequently, we compared the ability of the PAT graft and control groups to form granulation tissue in perichondrial defects. Granulation tissue is defined as connective tissue comprising new capillaries, spindle cells, mononuclear inflammatory cells, and neutrophils. A blinded pathologist measured granulation tissue formation in the wounds. The vertical and horizontal growth abilities of the granulation tissue were assessed by measuring the thickness and length of the tissue in the histological images stained with H&E. Each wound was divided into 2 mm segments, with the central 10-mm area designated as the perichondrial defect and the outer 10-mm area as the healthy perichondrium (Fig 1B). The segment with the greatest growth thickness in each region was analyzed. The thickness and length of the granulation tissue were measured using a Hamamatsu NanoZoomer digital pathology microscope (Hamamatsu Photonics, Hamamatsu City, Japan).

## 2.5 Wound contraction and epithelialization length

The region of the created wound (20 mm in length) was evaluated for open wounds, epithelialization, and wound contraction using vertical sectional H&E staining (Fig 1C). The open wound length refers to the portion of the wound that lacks an epidermis. In accordance with a previous methodology, the epithelialization length was determined as the distance from the

edge of the first hair follicle closest to the wound to the epithelial tip [13]. The wound contraction length was calculated as the difference between the length of the created wound (20 mm) and the distance between the two closest hair follicles. The length of wound closure was defined as the sum of the wound contraction and epithelialization lengths. All measurements were performed using a Hamamatsu NanoZoomer digital pathology microscope (Hamamatsu Photonics, Hamamatsu City, Japan).

## 2.6 Granulation tissue coverage rate

The coverage rate of granulation tissue over the cartilage was determined using the following formula:

$$\text{Coverage Rate} = \frac{\text{Length of granulation tissue}}{\text{Length of open wound}}$$

## 2.7 CD31+ cell counts and CD34+ and αSMA expression levels in the exposed cartilage

CD34+/CD31- is the biomarker for vascular progenitor/stem cells and endothelial cells are CD34+/CD31+ [14–16]. In this study, we assessed CD31+ cells as indicators of mature vessels, and CD34+ cells as cells with the potential to mature into vessels. Increased levels of αSMA are closely associated with enhanced wound contraction and skin contracture [17]. Hence, we evaluated αSMA levels as indicators of wound contraction and skin contracture. We analyzed granulation tissue to measure the percentage of tissue area positively stained for αSMA. Immunohistochemical images of CD31 and CD34 were captured using a NanoZoomer system at 10× magnification. Five fields corresponding to the segments defined in section 2.4 were randomly selected for perichondrium defects. The number of vessels stained with CD31+ is quantified in each image. For αSMA staining, five randomly chosen fields of granulation tissue were imaged at 40× magnification using the NanoZoomer system. The levels of CD34 and αSMA were evaluated using ImageJ and a previously published protocol [18].

## 2.8 Data-independent acquisition (DIA) mass spectrometry analysis

Quantitative DIA mass spectrometry was performed on rabbit PAT at the Kazusa DNA Research Institute (Kisarazu, Japan) [19]. Briefly, PAT (20 mg) was washed, solubilized, and ultrasonicated in acetonitrile supplemented with 0.1% trifluoroacetic acid. Subsequently, 0.5% of sodium dodecanoate was added to the solution. The samples were reduced using dithiothreitol, alkylated at the cysteine residues with iodoacetamide, and digested with Lys-C protease and trypsin. The resulting peptides were purified and analyzed by liquid chromatography-mass spectrometry using an UltiMate 3000 RSLC Nano LC System (Thermo Fisher Scientific, Waltham, MA, USA) and a Q Exactive HF-X mass spectrometer (Thermo Fisher Scientific). The detected mass spectrometry peaks were analyzed using the Scaffold DIA software ver. 3.2.1 (Proteome Software, Portland, OR, USA), with the UniProt rabbit proteome (Rabbit UniProtKB/Swiss-Prot database, UP000001811) as the basis for analysis. For Gene Ontology (GO) analysis, owing to the insufficient functional annotation of rabbit genes, rabbit genes were replaced with human genes using their corresponding gene symbols. Genes without annotations were excluded from analysis. The top 100 proteins identified in DIA proteome analysis were categorized into extracellular molecule (ECM) proteoglycans, non-proteoglycan ECMs,

other secreted proteins, plasma membrane, cytoplasm, and nucleus based on their annotations.

Furthermore, the top 100 proteins in PAT were subjected to GO term enrichment analysis using the Metascape web application (https://metascape.org/gp/index.html#/main/step1) to visualize the functional relationships among PAT proteins [20]. A recent study identified 18 proangiogenic extracellular components that initiate angiogenesis, and 12 antiangiogenic extracellular components that inhibit angiogenesis [21]. Based on this knowledge, we analyzed the extracellular components of PAT and identified both proangiogenic and antiangiogenic proteins.

### 2.9 Statistical analysis

All data were subjected to statistical analysis using the independent-samples t-test for normally distributed data and the Mann–Whitney U test for non-parametric data using SPSS software ver. 24.0 (IBM Corp., Armonk, NY, USA, 2016). Statistical significance was set at $P < 0.05$.

## 3. Results

In this study, we observed no adverse events that could negatively impact animal welfare.

### 3.1 Histological and immunohistochemical PAT analyses

The results of H&E and Elastica van Gieson staining revealed that the PAT comprised collagen fibers, abundant elastic fibers (blue arrow), and horizontally oriented capillary vessels (pink arrow). Some vessels ran parallel to the nerves (yellow arrow; Fig 2A, upper panel). In the vertical plane (Fig 2A, lower panel), the PAT could be divided into two regions: a membrane area composed of thin collagen fibers and a slightly bulging area containing vessels, which we designated the membrane and vascular areas. Upon magnification of the tissue within the vascular area, we observed vascular endothelial cells (pink arrow; CD31+), vascular progenitor/stem cells (orange arrow; CD34+/CD31) surrounding them, and nerves (yellow arrow; S100+) (Fig 2B, upper panel). The membrane comprises three to four sublayers of collagen, with numerous mononuclear and spindle cells evenly distributed throughout. These cells were identified as vascular progenitor/stem cells (orange arrow; CD34+/CD31; Fig 2B, lower panel).

### 3.2 Wound analysis

In the control group, granulation tissue formation was limited to the perichondrium on day 7 (Fig 2C, left and upper panel). Conversely, in the PAT group, a thin membrane or granulation tissue covered the exposed cartilage (Fig 2C, right and upper panel). In the control group, a narrower area of exposed cartilage and increased granulation tissue were observed on day 14. The wound surfaces exhibited a depression at the center (Fig 2C, left and lower panel). In contrast, the PAT group exhibited a flattened wound surface with no cartilage exposure (Fig 2C, right and lower panel).

### 3.3 PAT graft survival

On day 7, the PAT group exhibited a visible capillary vessel within a thin membrane that crossed the exposed cartilage (Fig 2D, above the left, indicated by yellow dotted lines). However, this effect was not observed in the control group. Histological and immunohistochemical examinations in the vertical plane revealed neurovascular bundles resembling the vascular region of the PAT in the perichondrial defect observed on days 7 and 14 (Fig 2D, below,

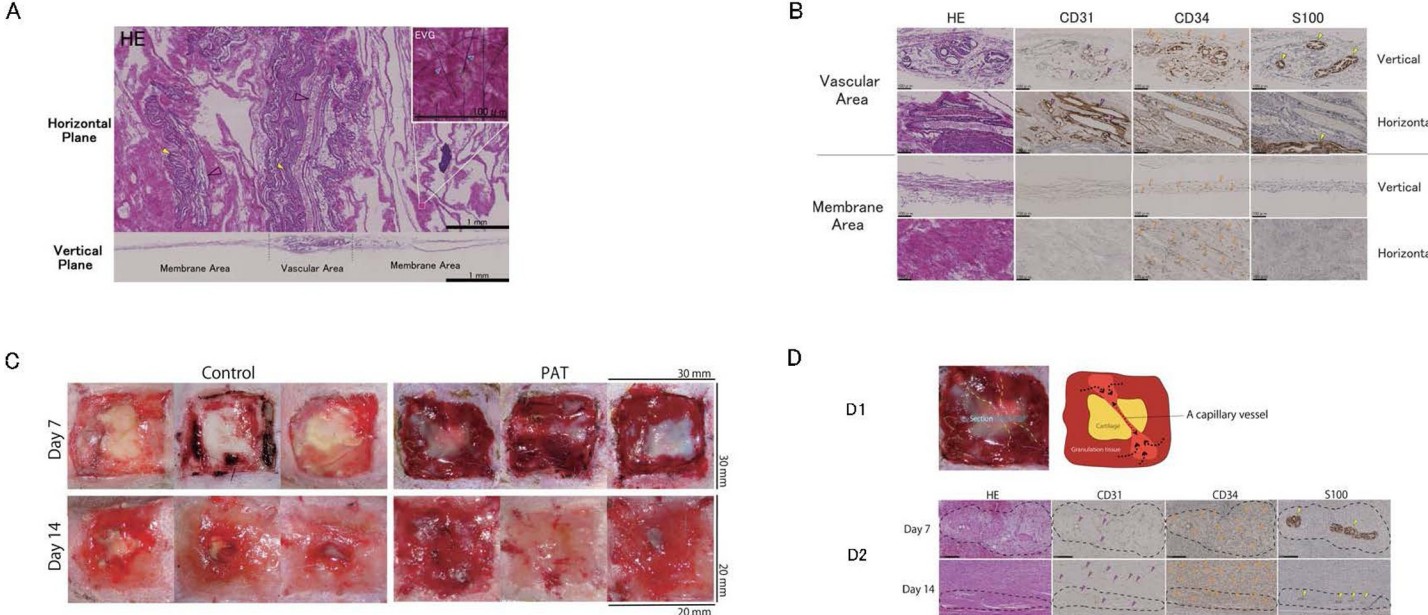

**Fig 2. Analysis of perifascial areolar tissue (PAT). A.** Hematoxylin and eosin (H&E) and Elastica van Gieson staining of PAT, with capillary vessels (pink arrow), nerves (yellow arrow), and elastic fibers (blue arrow) are indicated. **B.** Histological and biochemical evaluation of the cell layer in PAT under a high-magnification field, shown in the sequence of H&E staining, anti-CD31 immunostaining (CD31), anti-CD34 immunostaining (CD34), and anti-S100 immunostaining (S100). Vascular endothelial cells are denoted with pink arrows; orange arrows indicate vascular progenitor/stem cells; and yellow arrows indicate nerves. **C.** Representative macroscopic wound healing and histological findings. **D.** Macroscopic findings and histological and immunohistochemical staining of PAT survival. **D1**: vascular bridging on day 7 in macroscopic findings and illustration. Yellow dotted line: vascular bridging. **D2**: representative image of histological and immunohistochemical findings in vascular bridging at the perichondral defect in the vertical plane on days 7 and 14, shown in the sequence of H&E staining, anti-CD31 immunostaining (CD31), anti-CD34 immunostaining (CD34), and anti-S100 immunostaining (S100). The vascular area is indicated with a black dotted line; vascular endothelial cells are indicated with pink arrows; orange arrows indicate vascular progenitor/stem cells; and yellow arrows indicate nerves.

indicated by black dotted lines). The CD31+ and CD34+ areas were localized within the neurovascular bundle region on day 7 and had expanded from this area by day 14 (Fig 2D).

At 7 days post-application, vascular bridging was grossly visible in two and four of the 6 cases in the histological images. By day 14, granulation tissue had nearly enveloped all the exposed cartilage areas, obscuring the gross observation of vascular bridging, although vascular formation was present in all cases in the histological images.

## 3.4 Wound healing evaluation

**3.4.1 Granulation tissue growth.** Compared to granulation tissue formation within the wounds in the control group, the area without the perichondrium showed poor granulation tissue growth. Specifically, on days 7 and 14, the area with a perichondrial defect exhibited significantly thinner and shorter granulation tissue than did the residual perichondrium (** $P < 0.001$ and * $P = 0.041$, respectively; Fig 3A). Regarding granulation tissue formation, it was revealed that only a small amount of granulation tissue was formed on the outer edge of the perichondrial defect on day 7 (Fig 3B, left and upper panel). By day 14, a narrower area of exposed cartilage and increased granulation at the peripheral area was observed (Fig 3B, left and lower panel). In contrast, the PAT group exhibited granulation tissue formation even at the center of the perichondrial defect on day 7 (Fig 3B, right and upper panel). A few cells were observed in areas devoid of granulation tissue; however, membranous structures were discernible. By day 14, significant granulation tissue growth was observed with no cartilage exposure on day 14 (Fig 3B, right and lower panel).

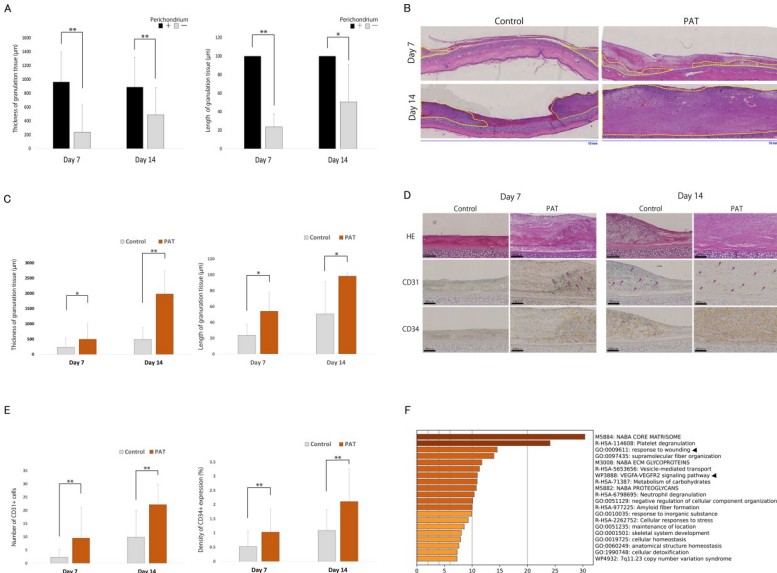

**Fig 3. Wound healing evaluation. A.** Comparison of granulation ability between perichondrium and the perichondrial defect areas in the control group. **Left:** thickness of granulation tissue. ** $P < 0.01$. N = 30. **Right:** length of granulation tissue. ** $P < 0.01$. N = 6. **B.** Hematoxylin and eosin (H&E) staining findings at the perichondrial defect on days 7 and 14. Yellow dotted line: granulation tissue. PAT: perifascial areolar tissue. **C.** Comparison of the control and PAT granulation ability at the perichondrial defect. **Left:** thickness of granulation tissue. * $P < 0.05$, ** $P < 0.01$. N = 30. **Right:** length of granulation tissue. * $P < 0.05$, ** $P < 0.01$. N = 6. **D.** Representative histological and immunohistochemical findings for angiogenesis at the perichondrial defect on days 7 and 14, shown in the sequence of H&E staining, anti-CD31 immunostaining (CD31), and anti-CD34 immunostaining (CD34). Red arrow: CD31+ cells; yellow arrow: CD34+/CD31 cells. **E.** Angiogenesis at the perichondrial defect. **Left:** count of CD31 expressing cells. **Right:** level of CD34 expression at the perichondrial defect. ** $P < 0.01$; N = 30. **F.** Gene Ontology term enrichment analysis of the top 100 molecules using Metascape. The upper black arrow indicates the response wound signaling pathway, and the lower black arrow indicates the vascular endothelial growth factor-A (VEGFA)/ vascular endothelial growth factor receptor-2 (VEGFR2) signaling pathway.

On day 7, the PAT group exhibited significantly thicker and longer granulation tissue over the perichondrial defect than the control group (* $P = 0.021$ and 0.041, respectively; Fig 3C). This trend was also observed on day 14 (** $P < 0.001$ and * $P = 0.046$ for thickness and length, respectively; Fig 3C). Furthermore, the coverage rate of granulation tissue over the cartilage in the PAT group was significantly higher than that in the control group on days 7 and 14 (* $P = 0.039$ and 0.036, respectively).

**3.4.2 Angiogenesis.** During histological and immunohistochemical analyses, the central perichondral defect images were evaluated on days 7 and 14 to assess angiogenesis. CD34 + cells were classified into CD31+ vascular endothelial cells (red arrows) and CD31 vascular progenitor/stem cells (yellow arrows). On day 7, neither CD31+ nor CD34+ cells were observed in the control group (Fig 3D, left-column panels). In the PAT group, on day 7, we observed thin collagen tissue containing CD34+/CD31 cells, resembling the membrane area of PAT, as well as granulation tissue containing CD31+ and CD34+ cells (Fig 3D, center left-column panels). On day 14, granulation tissue containing CD31+ and CD34+ cells formed and extended gradually towards the center of the wound in the control group (Fig 3D, center right-column panels). On day 14 in the PAT group, the granulation tissue exhibited significant growth and contained a uniform distribution of numerous CD31+ and CD34+ cells (Fig 3D, right-column panels). The number of CD31+ cells stained in the perichondral defect region of the PAT group was significantly higher than that of the control group on days 7 and 14 ($P < 0.01$) (Fig 3E, left). Additionally, the average relative abundance of CD34+ cells in the

PAT group was significantly higher than that in the control group on days 7 and 14 ($P < 0.01$) (Fig 3E, right).

GO term enrichment analysis of the top 100 molecules demonstrated a strong correlation with terms associated with the response to the wound signaling pathway (upper black arrow) and the VEGFA/VEGFR2 signaling pathway (lower black arrow) (Fig 3F).

The extracellular components of PAT consisted of 15 pro-angiogenic and two anti-angiogenic proteins (Table 1). The mass of pro-angiogenic molecules in PAT was 700 times greater than that of anti-angiogenic molecules. Numerous pro-angiogenic molecules were identified among the top 25 extracellular molecules of PAT, including fibrillin 1, decorin, collagen type 1, fibronectin 1, and collagen type 3. Furthermore, the vascular stem cell marker CD34 was identified among the top 25 hits in the plasma membrane. Three angiogenesis-promoting growth factors were identified among the top six growth factors: heparin-binding growth factor (HDGF), fibroblast growth factor 2 (FGF2), and platelet-derived growth factor D (PDGFD) (Table 2).

**3.4.3 Wound contraction evaluation.** The open wound area, as determined from the photographs, was significantly greater in the PAT group than in the control group on day 7 ($P < 0.05$) (Fig 4A). However, there was no significant difference between the groups on day 14 ($P = 0.16$). Similarly, the open wound length measured from the tissue sections was significantly longer in the PAT group than in the control group ($P = 0.004$) (Fig 4B). Furthermore, no significant differences were observed on day 14 ($P = 1.0$). These results indicated an initial delay in wound closure observed on day 7 in the PAT group. Subsequently, the wound closure rate increased and was comparable to that of the control group by day 14. Wound contraction was significantly shorter in the PAT group than in the control group on day 7 ($P < 0.01$) (Fig 3C). However, no significant differences were observed on day 14 ($P = 0.39$). The length of epithelialization was not significantly different between PAT and the control groups on days 7 and 14 ($P = 1.0$ and 0.093, respectively). The rates of epithelialization and wound contraction

**Table 1. Extracellular molecules and fragments associated with vessel formation exhibiting proangiogenic and antiangiogenic activities.**

| Activity | Number | Gene name | Protein name | Identified peptide count | Protein cmount |
|---|---|---|---|---|---|
| Proangiogenic | 1 | FBN1 | Fibrillin 1 | 62 | 8.05E+09 |
| | 2 | DCN | Decorin | 10 | 2.04E+09 |
| | 3 | COL1A1 | Collagen alpha-1(I) chain | 6 | 7.59E+08 |
| | 4 | FGB | Fibrinogen beta chain | 21 | 3.85E+08 |
| | 5 | FGA | Fibrinogen alpha chain | 16 | 3.59E+08 |
| | 6 | COL1A2 | Collagen alpha-2(I) chain | 11 | 3.24E+08 |
| | 7 | FGG | Fibrinogen gamma chain | 14 | 2.83E+08 |
| | 8 | FN1 | Fibronectin | 30 | 1.57E+08 |
| | 9 | COL3A1 | Collagen type III alpha 1 chain | 9 | 1.18E+08 |
| | 10 | LAMC1 | Laminin subunit gamma 1 | 15 | 4.41E+07 |
| | 11 | VTN | Vitronectin | 5 | 3.85E+07 |
| | 12 | COL4A2 | Collagen type IV alpha 2 chain | 5 | 2.17E+07 |
| | 13 | LAMB1 | Laminin subunit beta 1 | 13 | 1.94E+07 |
| | 14 | COL15A1 | Collagen type XV alpha 1 chain | 6 | 1.72E+07 |
| | 15 | COL4A1 | Collagen type IV alpha 1 chain | 2 | 1.05E+07 |
| Antiangiogenic | 1 | THBS1 | Thrombospondin 1 | 10 | 1.40E+07 |
| | 2 | - | Endostatin domain-containing protein | 3 | 4.05E+06 |

The ratio of antiangiogenic to proangiogenic molecules = 700

**Table 2. Top 25 extracellular matrix proteins, top 25 membrane proteins, and top six growth factors and cytokines in perifascial areolar tissue.**

| Categories | Gene name | Protein name | Identified peptide | Protein amount |
|---|---|---|---|---|
| Extracellular matrix | FBN1 | fibrillin 1 | 62 | 8.05E+09 |
| (top 25) | LUM | lumican | 9 | 5.07E+09 |
| | DCN | decorin | 10 | 2.04E+09 |
| | PRELP | proline and arginine rich end leucine rich repeat protein | 14 | 1.81E+09 |
| | COL14A1 | collagen type XIV alpha 1 chain | 37 | 1.72E+09 |
| | OGN | osteoglycin | 6 | 1.69E+09 |
| | POSTN | periostin | 26 | 1.11E+09 |
| | DPT | dermatopontin | 5 | 8.88E+08 |
| | COL1A1 | collagen type I alpha 1 chain | 6 | 7.59E+08 |
| | FBN2 | fibrillin 2 | 14 | 3.35E+08 |
| | COL1A2 | collagen type I alpha 2 chain | 11 | 3.24E+08 |
| | MBP | myelin basic protein | 6 | 3.03E+08 |
| | COL6A6 | collagen type VI alpha 6 chain | 34 | 3.00E+08 |
| | LGALS1 | galectin 1 | 4 | 2.89E+08 |
| | COL21A1 | collagen type XXI alpha 1 chain | 3 | 2.02E+08 |
| | MFAP5 | microfibril associated protein 5 | 3 | 1.71E+08 |
| | HSPG2 | heparan sulfate proteoglycan 2 | 48 | 1.67E+08 |
| | BGN | biglycan | 7 | 1.61E+08 |
| | FN1 | fibronectin 1 | 30 | 1.57E+08 |
| | COL2A1 | collagen type II alpha 1 chain | 2 | 1.28E+08 |
| | COL3A1 | collagen type III alpha 1 chain | 9 | 1.18E+08 |
| | TTR | transthyretin | 2 | 1.06E+08 |
| | TGFBI | transforming growth factor beta induced | 10 | 1.06E+08 |
| | LCN2 | lipocalin 2 | 5 | 1.05E+08 |
| | LAMB2 | laminin subunit beta 2 | 21 | 9.26E+07 |
| Plasma membrane | MPZ | myelin protein zero | 8 | 1.50E+09 |
| (top 25) | SPTAN1 | spectrin alpha, non-erythrocytic 1 | 85 | 6.21E+08 |
| | SPTBN1 | spectrin beta, non-erythrocytic 1 | 68 | 4.51E+08 |
| | ANXA5 | annexin A5 | 10 | 4.02E+08 |
| | CLTC | clathrin heavy chain | 38 | 1.99E+08 |
| | ANXA6 | annexin A6 | 18 | 1.86E+08 |
| | VCL | vinculin | 28 | 1.37E+08 |
| | TLN1 | talin 1 | 48 | 1.14E+08 |
| | THY1 = CD90 | Thy-1 cell surface antigen | 3 | 1.04E+08 |
| | ANXA1 | annexin A1 | 12 | 8.80E+07 |
| | SLC4A1 | solute carrier family 4 member 1 (Diego blood group) | 12 | 8.52E+07 |
| | ATP1A1 | ATPase Na+/K+ transporting subunit alpha 1 | 19 | 8.39E+07 |
| | PLP1 | proteolipid protein 1 | 5 | 7.49E+07 |
| | PRPH | peripherin | 11 | 6.85E+07 |
| | AOC3 | amine oxidase copper containing 3 | 9 | 6.54E+07 |
| | GNA13 | G protein subunit alpha 13 | 3 | 6.34E+07 |
| | SPTB | spectrin beta, erythrocytic | 38 | 6.21E+07 |
| | ACTR2 | actin related protein 2 | 8 | 5.24E+07 |
| | ANXA4 | annexin A4 | 11 | 4.74E+07 |
| | EHD4 | EH domain containing 4 | 15 | 4.49E+07 |
| | ANXA8/ANXA8L1 | annexin A8 like 1 | 12 | 4.05E+07 |
| | UTRN | utrophin | 37 | 3.53E+07 |

*(Continued)*

**Table 2.** (Continued)

| Categories | Gene name | Protein name | Identified peptide | Protein amount |
|---|---|---|---|---|
| | CD34 | CD34 molecule | 4 | 3.36E+07 |
| | ACTR3 | actin related protein 3 | 7 | 3.31E+07 |
| | NEFM | neurofilament medium | 8 | 3.21E+07 |
| Growth factors | HDGF | heparin binding growth factor | 6 | 7.26E+06 |
| | FGF2 | fibroblast growth factor 2 | 1 | 2.98E+06 |
| | AGT | angiotensinogen | 3 | 1.33E+06 |
| | PDGFD | platelet derived growth factor D | 1 | 3.06E+05 |
| | GRN | granulin precursor | 1 | 1.50E+05 |
| | MST1 | macrophage stimulating 1 | 1 | 1.48E+05 |
| Cytokines | MIF | macrophage migration inhibitory factor | 1 | 1.63E+07 |
| | C5 | complement C5 | 8 | 1.10E+07 |
| | AIMP1 | aminoacyl tRNA synthetase complex interacting | 4 | 3.79E+06 |
| | | multifunctional protein 1 | | |
| | IL18 | interleukin 18 | 1 | 3.06E+06 |
| | WNT2 | Wnt family member 2 | 4 | 1.48E+06 |
| | FAM3C | FAM3 metabolism regulating signaling molecule C | 1 | 7.27E+05 |

relative to the initial wound length are shown in Fig 4D. In the control group, wound closure was 18% on day 7 (6% owing to epithelialization and 12% owing to wound contraction). On day 14, wound closure increased to 37% (17% epithelialization and 20% wound contraction), with a greater contribution from wound contraction. Conversely, in the PAT group, wound closure was 6% on day 7 (5% epithelialization and 1% wound contraction) and 41% on day 14 (25% epithelialization and 16% wound contraction), indicating a higher rate of wound closure through epithelialization. The expression of αSMA in the PAT group was less prominent than in the control group on day 7 (Fig 4E). The average relative level of αSMA in the PAT group was significantly lower than that in the control group on day 7 ($P < 0.01$) (Fig 4F).

## 4. Discussion

### 4.1 Healing wounds with ischemic structure exposure

**4.1.1 Delayed healing of wounds with ischemic structure exposure.** Defects in the perichondrium exhibited significantly impaired granulation ability compared with intact perichondrium areas. Furthermore, we observed that vascular granulation tissue formed gradually in a horizontal direction from the vascular region towards the ischemic part of the wound, with inadequate granulation tissue formation at the central wound site. These findings suggest the potential for elucidating the challenges of healing wounds with exposed ischemic structures.

**4.1.2 Healing of wounds with ischemic structure exposure treated with PAT grafts.** PAT grafts were superior to occlusive dressings in covering exposed ischemic structures with granulation tissue and promoting angiogenesis. By day 7, the PAT graft had formed granulation tissue in a portion of the perichondrial defect, with continued vertical and horizontal growth until day 14. This enhanced wound healing can be attributed to vascular bridging and the local administration of cells and molecules that promote angiogenesis.

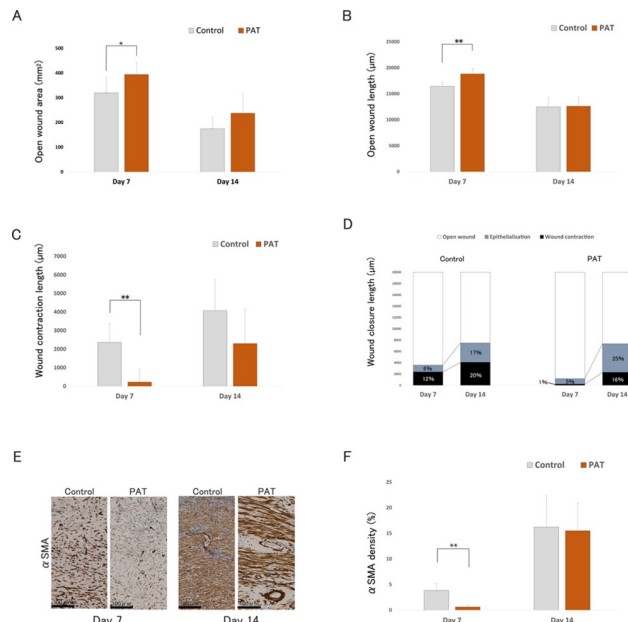

**Fig 4. Wound closure analysis. A.** Open wound area in each group on days 7 and 14. * $P < 0.05$. N = 6 per group. **B.** Open wound length of each group on days 7 and 14. ** $P < 0.01$. N = 6 per group. **C.** Wound contraction length of each group on days 7 and 14. ** $P < 0.01$. N = 6 per group. **D.** Wound closure length, rate of epithelialization, and wound contraction. The displayed value represents the rate of epithelialization and wound contraction lengths compared to the created wound length. **Left:** control. **Right:** PAT. **E.** Anti-α-SMA immunostaining. **Left:** control on day 7; second: PAT on day 7; third: control on day 14; **Right:** PAT on day 14. **F.** Level of α-SMA expression on days 7 and 14. ** $P < 0.01$. N = 30 per group. PAT: perifascial areolar tissue group.

## 4.2 Vascular bridging

Previous clinical studies have speculated that the survival of PAT grafts is facilitated by capillary anastomosis between the capillaries in the PAT and those in the recipient wound bed [3, 5, 6]. However, no studies have provided definitive evidence of vascular bridging. In the present study, we observed capillaries crossing the cartilage and identified vascular nerve bundles in the grafted PAT over ischemic tissue. These findings support the hypothesis that vascular bridging does indeed occur. Vascular bridging is analogous to the bridging phenomenon caused by skin grafting on ischemic tissue. The most recent research on the bridging phenomenon dates back to the 1970s and the 1980s, leaving the underlying principles to be fully elucidated [22, 23]. Studies in rabbits, where skin grafts were applied to exposed wounds of avascular structures, revealed that the bridging phenomenon occurred in full-thickness skin grafts, but not in split-thickness skin grafts [22, 23]. Sprouting angiogenesis from the graft bed and its connection to the native graft vasculature is necessary to survive full-thickness skin grafts [24–26]. After connection, blood flows through the native graft vasculature, and recipient-derived vascular progenitor cells reconstruct the vascular network. Because the vessels sprouting from the graft bed are limited to vertical extension, the graft must have a horizontal native vasculature for vascularization over the ischemic tissue. The inherent thickness of full-thickness skin grafts allows for their conceptualization as horizontally vascularized in a three-dimensional context. Furthermore, full-thickness skin grafts have thicker and longer capillaries, facilitate bridging with minimal vascular linkage [27]. Thus, the histological feature of successful vascular bridging could be a horizontal vascular architecture with long and thick capillaries. Comparing the PAT in this study with full-thickness skin grafts in previous studies [27], PAT had these features more strongly.

### 4.3 Provision of factors to promote wound healing and angiogenesis

The presence of progenitor/stem cells and various molecules within PAT grafts that improve wound healing may promote granulation tissue formation in ischemic tissues.

### 4.4 CD34+ administration

Previous studies have demonstrated the wound-healing benefits of locally administered CD34 + cells. In a study involving mice with diabetes, human endothelial progenitor cells (CD34 + cells) were injected locally into the wound bed, which led to neovascularization in conjunction with recipient endothelial cells, increased wound vascular density, and a high rate of wound closure [28]. A recent clinical trial showed that the local administration of human peripheral blood CD34+ cells promoted the healing of diabetic foot ulcers [29]. Thus, promotion of wound healing and angiogenesis in ischemic tissues by PAT is likely related to the local administration of abundant CD34+ cells within PAT. However, it should be noted that our findings did not confirm the uniform presence of CD34+ cells throughout the graft, suggesting that some of the administered CD34+ cells did not survive. Previous studies have reported that when transplanted as free fat graft intra-tissues, CD34+ cells, including adipose stem cells, can be divided into surviving and apoptotic populations, depending on the blood supply after grafting [30, 31]. Surviving cells play an indirect paracrine role in response to the apoptosis of other cells, triggering the movement of recipient-derived CD34+ cells and facilitating the infiltration of host vessels into the graft [30, 31]. A similar mechanism may occur with PAT grafts.

### 4.5 Extracellular matrix and growth factor/cytokine administration

The significant enrichment of molecules associated with the wound response and VEGF signaling in the ECMs of PAT strongly suggests that PAT serves as a platform for promoting wound healing and enhancing angiogenesis.

Fibrillin 1 and decorin, which promote angiogenesis [32–34] accounted for 10% of the total mass of the ECM and the extreme weight gradient of proangiogenic molecules relative to anti-angiogenic molecules in the ECM. In addition, HDGF, FGF2, and PDGF, which are abundant in PAT, are growth factors that promote angiogenesis and granulation tissue formation, likely explaining the enhanced angiogenesis observed in the current study [35–38]. Furthermore, considering the roles of macrophage migration inhibitory factor (MIF), complement C5, and aminoacyl-tRNA synthetase-interacting multi-functional protein 1 (AIMP1), present in abundance in PAT, it is plausible that extracellular cytokines promoting granulation, cell proliferation, and wound healing contribute to the apparent effectiveness of PAT grafts [39–42].

This study identified numerous molecules within PAT that promote wound healing and angiogenesis. However, the precise mechanisms underlying their functions have not been fully explored. Insights into these mechanisms may be gleaned from decellularized ECM membrane biomaterials, such as the acellular dermal matrix, small intestinal submucosa, and acellular amnion. These materials are widely utilized for skin wound healing, retain various bioactive substances including growth factors, collagen, laminin, fibronectin, and polysaccharides [43]. Although the exact wound repair mechanisms in living organisms remain to be fully understood, the benefits of these materials for wound healing are hypothesized to stem from the following: 1) their thin, flat scaffold structure is conducive to high-density cell seeding and migration of reparative cells from adjacent tissues, and 2) the provision of a complex signaling milieu on the wound surface via the extracellular matrix, growth factors, and cytokines, which collectively stimulate granulation tissue formation, angiogenesis, and matrix remodeling [43]. Given the structural and compositional similarities of PAT ECM to these membranous biomaterials, including membrane structure, the presence of various extracellular matrix

components, and the inclusion of growth factors and cytokines, it is reasonable to posit that the mechanism of action of PAT grafts as scaffolds for wound healing mirrors that of these materials.

## 4.6 Wound contraction control with PAT graft

Skin wounds undergo closure through a combination of epithelialization (epithelial migration from the wound edge) and contraction (bringing the wound edges closer together) [44]. Wound contraction is crucial for creating a mechanically robust scar, reducing the wound area, and promoting healing. However, in humans, excessive contractions can lead to hypertrophic scarring, resulting in cosmetic and functional issues [45]. Therefore, there is a need for transplant materials with greater reepithelialization and reduced wound contraction.

In the present study, the application of PAT grafts suppressed wound contraction and promoted epithelialization without prolonging the overall wound closure timeline. Additionally, it induced lower levels of α-SMA expression in wound contracture and scarring [17, 45–47]. Consequently, our findings suggest that PAT grafts can be used as a transplant material to mitigate excessive wound contraction and the development of hypertrophic scarring.

Decorin, an extracellular protein found at relatively high levels in PAT, may suppress wound contraction. Decorin is a crucial proteoglycan in the normal dermis [48]. Zhang et al. [48] suggested that it can potentially play a role in preventing and treating excessive skin contraction in hypertrophic scarring, as recombinant human decorin inhibits cell proliferation and downregulates TGF-β1 production in hypertrophic scar fibroblasts. Our previous study, which involved administering human recombinant decorin in mouse wounds, yielded results similar to those of the present study, specifically, the inhibition of early wound contraction, the promotion of epithelialization, and the down-regulation of α-SMA [49].

## 4.7 Mechanisms underlying PAT graft viability

In summary, the wound healing process facilitated by PAT grafts appears to proceed as follows. Vascular bridging occurs through the integration of neovascular vessels from the PAT graft with those in the graft bed. Some CD34+/CD31 progenitor/stem cells present in the membrane area of the PAT are removed through ischemia after grafting. The other surviving cells induce the migration of recipient-derived stem cells through an indirect paracrine mechanism. These induced cells spread throughout the transplanted tissue through the horizontal vascular structure of PAT and promote angiogenesis, cell proliferation, and differentiation, forming granulation tissue on the exposed ischemic tissue. Furthermore, the scaffolding properties of PAT, which are conducive to cell seeding and migration, along with the ECM, growth factors, and cytokines collectively promote granulation tissue formation, angiogenesis, matrix formation, and remodeling. Fig 5 provides a concise visual representation of the wound-healing process involving the integration of PAT grafts into ischemic structures.

## 4.8 Comparative efficacy of PAT grafting versus other treatments

**4.8.1 Distinctive aspects relative to skin grafting.** Reports suggest that full-thickness grafts are clinically viable when ischemic tissue exposure is limited to 1.5 cm or less [23]. However, these findings lacked robust evidence-based confirmation. Furthermore, studies involving full-thickness skin grafts on exposed wounds of avascularized structures in rabbits indicate that although survival of such grafts is possible, their success rates are variable [22, 23]. The current consensus discourages the use of skin grafts for wounds with exposed tendons or bones. In contrast, in our previous clinical study, PAT grafts survived completely in tendon-

## PAT graft

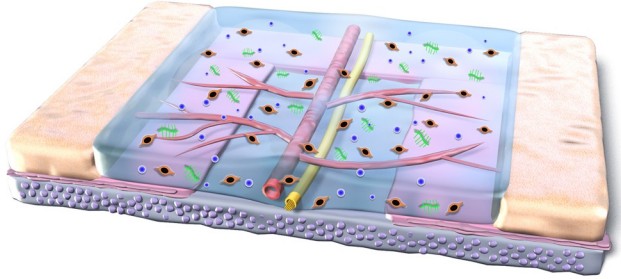

## Vascular Bridging

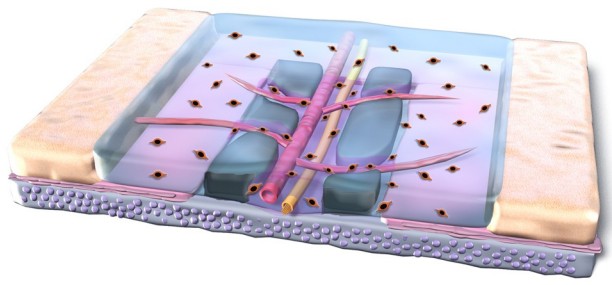

## Vascular tissue expanding

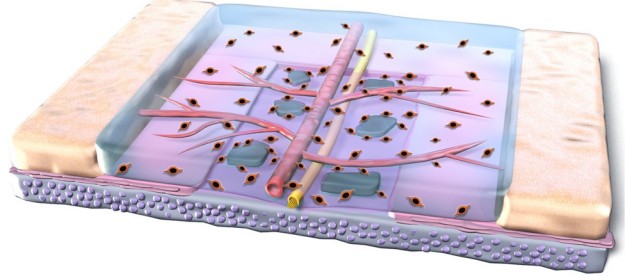

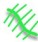 ECM
(fibrillin 1, decorin, collagen type1, etc.)

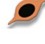 Progenitor/stem cells
(CD34+CD31− cells)

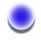 Growth factors/cytokines
(HDGF, FGF2, PDGFD, etc)

**Fig 5. Summary illustration of the PAT graft in this study.** The upper graphic depicts the depiction is of the immediate aftermath of grafting. PAT includes a vascular plexus with neurovascular bundles, CD34+/CD31 cells, various growth factors/cytokines, and extracellular matrix molecules grafted on a full-thickness skin defect in the center of the perichondral defect. The middle graphic illustrates the process of vascular bridging observed on day 7 post-grafting. The lower graphic portrays the expansion of vascular tissue noted on day 14. PAT: perifascial areolar tissue; ECM: extracellular molecule; HDGF: heparin-binding growth factor; FGF2: fibroblast growth factor 2; PDGFD: platelet-derived growth factor D.

exposed wounds of up to 3 cm in width [5]. The apparent survival of PAT grafts compared to skin grafts for ischemic tissue exposure wounds can be attributed to the following reasons:

1. Split-thickness skin grafts generally survive easily because of their ischemic tolerance. The ischemic tolerance is made possible by the thinness of the graft, which allows for longer periods to await vascular connection between the recipient wounds and grafts [27]. However, they are less viable in ischemic tissues because of the absence of a well-developed horizontal vascular network [22].

2. The vascular structure of full-thickness skin grafts is suitable for wounds with exposed ischemic tissue. However, they have less ischemic tolerance because the volume of the tissue is high.

3. Conversely, PAT grafts exhibit distinct benefits in terms of ischemic tolerance and vascular structure, which are attributed to their thin tissue and horizontal vascular network with long, thick individual vessels. Moreover, the membranous and sparse connective tissue composition of the PAT may facilitate cell migration and seeding, thereby enhancing granulation tissue formation [43].

**4.8.2 Comparative analysis with other treatment modalities.** Recent studies have highlighted the efficacy of negative pressure wound therapy (NPWT) and biomaterials in treating wounds with exposed ischemic tissue.

NPWT promotes granulation in tissues with adequate blood circulation. However, as granulation tissue cannot form directly on the ischemic tissue, the mechanism of wound closure under NPWT mirrors that observed in the control groups in this study is primarily dependent on wound contraction [50]. Furthermore, clinical applications of negative pressure closure therapy generally entail longer treatment durations than those entailed by PAT grafts [5], suggesting that PAT grafts offer benefits in terms of reduced treatment time and minimized scar contracture.

The role of scaffolding and the contribution of ECM molecules, cytokines, and growth factors in promoting wound healing are similar to those of biomaterials [43]. However, these biomaterials do not facilitate vascular bridging, which highlights the superiority of PAT grafts. Moreover, such biomaterials are typically devoid of cells and, therefore, cannot deliver stem cells directly to the wound site. PAT grafts are superior in terms of vascular bridging and local administration of progenitor/stem cells. This capability is pivotal, as even in the absence of vascular bridging, PAT grafts can still contribute remarkably to granulation tissue formation owing to their inherent stem cell content.

One of the limitations of this study was its small sample size. Additionally, using a rabbit model may introduce differences compared to human PAT. To address these limitations, future studies should focus on the efficacy of PAT grafts in humans.

## 5. Conclusions

This is the first report to quantitatively demonstrate the therapeutic efficacy of PAT grafts and highlight their utility in treating wounds with exposed ischemic structures. The effectiveness of PAT grafts can be attributed to two primary factors: vascular bridging during graft survival and the provision of three essential elements that promote wound healing and angiogenesis (progenitor/stem cells, ECM molecules, and growth factors/cytokines). In addition, PAT grafts can serve as transplantable tissues that inhibit scar contraction.

## Supporting information

**S1 Table. Antigen activation and dilutions and incubation times for primary antibodies.**
(XLSX)

**S1 Dataset. Minimal underlying data set for the study.**
(XLSX)

## Acknowledgments

The authors thank the members of the Department of Plastic Surgery, Biochemistry, and Pathology at Kanazawa Medical University for their valuable contributions. Special thanks to H. Yonekura for overseeing the molecular mass spectrometry and Y. Ueda for supervising the histological and immunohistochemical analyses. We would like to thank Editage (www.edi-tage.jp) for English language editing.

## Author Contributions

**Conceptualization:** Toru Miyanaga.

**Data curation:** Toru Miyanaga, Yasuo Yoshitomi.

**Formal analysis:** Toru Miyanaga, Yasuo Yoshitomi.

**Funding acquisition:** Toru Miyanaga.

**Investigation:** Toru Miyanaga, Yasuo Yoshitomi, Aiko Miyanaga.

**Project administration:** Toru Miyanaga.

**Resources:** Toru Miyanaga.

**Supervision:** Toru Miyanaga.

**Visualization:** Toru Miyanaga.

**Writing – original draft:** Toru Miyanaga.

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
