## [Decision Letter · Decision Letter 0]

23 Oct 2023

PONE-D-23-30878Perifascial areolar tissue graft promotes angiogenesis and wound healing in exposed ischemic component rabbit modelPLOS ONE

Dear Dr. Miyanaga,

Thank you for submitting your manuscript to PLOS ONE. After careful consideration, we feel that it has merit but does not fully meet PLOS ONE’s publication criteria as it currently stands. Therefore, we invite you to submit a revised version of the manuscript that addresses the points raised during the review process.

The study is interesting and the manuscript is well written; however, it requires minor revisions. Please answer all questions posed by reviewers.

We look forward to receiving your revised manuscript.

Kind regards,

Alessandra Giuliani

Academic Editor

PLOS ONE

Journal Requirements:

   "This research received support from the Japan Society for the Promotion of Science (JSPS) KAKENHI Grant Number 19K12809 and an SBC Research Support Grant."

5. We notice that your supplementary table 1 are included in the manuscript file. Please remove them and upload them with the file type 'Supporting Information'. Please ensure that each Supporting Information file has a legend listed in the manuscript after the references list.

Additional Editor Comments:

Dear Authors,

In this study you demonstrated the effectiveness of PAT grafts and their therapeutic and potential impact on clinical practice. The study is interesting and the manuscript is well written. However, it requires minor revisions. Please answer all questions posed by reviewers.

Reviewers' comments:

Reviewer's Responses to Questions

**Comments to the Author**

1. Is the manuscript technically sound, and do the data support the conclusions?

Reviewer #1: Yes

Reviewer #2: Yes

2. Has the statistical analysis been performed appropriately and rigorously? 

Reviewer #1: Yes

Reviewer #2: Yes

3. Have the authors made all data underlying the findings in their manuscript fully available?

Reviewer #1: Yes

Reviewer #2: Yes

4. Is the manuscript presented in an intelligible fashion and written in standard English?

Reviewer #1: Yes

Reviewer #2: Yes

5. Review Comments to the Author

Reviewer #1: The authors demonstrated in this study the efficacy of PAT grafts and its therapeutic and potential impact on clinical practice. The study is interesting and the manuscript is well written.

Minor revisions:

1. Figure 2B: both in the figure and in the text (line 241) the magnification and scale bars lack

2. Line 287- line 291: P has to be connected with asterisks

3. please amplify the discussion pointing out the advantages of this methods (PAT) compared to the others

Reviewer #2: In the name of God

Dear Author

Hello,

I read your interesting manuscript. My questions, comments and questions are as follows:

1. I found a few grammatical errors.

2. Why perifascial areolar tissue (PAT) grafts is effective to promote granulation tissue formation that other tissues such as skin graft or pure fat doesn’t.

3. Did you consider PAT viability after grafting on an bare area?

4. What’s the mechanism of action of PAT for granulation tissue formation, it’s effect as a viable graft or it’s effect when necrosis occurs and intra cellular mediators such as growth factors release? If first hypothesis is correct please mention what’s the mechanism of PTA viability.

5. You stated: “The open wound area, as determined from photographs, was significantly larger in the PAT 345 group than in the control group on day 7 (P < 0.05) (Fig 4A)”. Does this mean wounds on both groups were not similar?

6. What’s the mechanism of viability of PAT graft.

7. You stated “The effectiveness of PAT grafts can be attributed to two primary factors, namely vascular bridging during graft survival and the provision of three essential elements that promote wound healing and angiogenesis: progenitor/stem cells, ECM molecules, and growth factors/cytokines. ”. My questions is why skin graft cannot do this? To best of our knowledge skin graft cannot cover or take on bare areas such as bone without periosteum but you concluded PAT can. Please clearly stat what’s the difference.

8. I think the manuscript is too long and includes parts that are unnecessary.

Regards

6. PLOS authors have the option to publish the peer review history of their article (what does this mean?). If published, this will include your full peer review and any attached files.

Reviewer #1: No

Reviewer #2: No

---

## [Author Response · Author response to Decision Letter 0]

15 Dec 2023

Response to the comments of the Reviewer 

Dear Editor,

Thank you for evaluating our manuscript, your appreciation of our work, and your encouraging comments. We appreciate your comments and the reviewers’ guidance and thoughtful suggestions from which the manuscript has benefited. We have read the comments provided by you and the reviewers and have revised the manuscript accordingly. Please find our point-by-point responses to all comments below. 

Reviewer #1: 

1. Figure 2B: both in the figure and in the text (line 241) the magnification and scale bars lack.

(Response)

Thank you for pointing this out. As such, the appropriate magnification levels and scale bars have been integrated into the Figure 2B.

2. Line 287- line 291: P has to be connected with asterisks

(Response)

We apologize for this oversight. The missing asterisks have been inserted.

3. Please amplify the discussion pointing out the advantages of this

methods (PAT) compared to the others

(Response)

We appreciate your insightful recommendation. We have expanded our discussion to emphasize the benefits of the PAT method over alternative approaches. Under the new subheading “Comparative efficacy of PAT grafting versus other treatments” in section 4.8, we provide a comprehensive analysis, comparing PAT grafts with other clinically employed treatments for ischemic tissue-exposed wounds.

Reviewer #2: 

There are some concerns:

1. I found a few grammatical errors.

(Response)

We have submitted the manuscript for an additional round of revision with Editage to ensure that all grammatical errors are corrected.

2. Why perifascial areolar tissue (PAT) grafts is effective to promote granulation tissue formation that other tissues such as skin graft or pure fat doesn't.

(Response)

We appreciate your query. We have now incorporated a detailed comparison with skin grafting, as suggested. Current understanding acknowledges that full-thickness skin grafts have the potential for integration with ischemic tissues. However, the specific mechanisms facilitating this process remain inadequately elucidated. Furthermore, the consensus is that such grafts are typically not indicated for application on ischemic tissues. We have reviewed in detail the reports on the methodology of skin grafting and incorporated them into our discussion, particularly emphasizing this in the “Vascular bridging” subheading in Section 4.2. It should be noted that our discussion does not encompass fat grafting, as this technique is predominantly applied to intact tissue rather than to open wounds.

3. Did you consider PAT viability after grafting on a bare area? 

(Response)

We appreciate your invaluable suggestion. Our observations revealed that at 7 days post-application, vascular bridging was grossly visible in two of the six cases and four of the six cases on histological analysis. By day 14, granulation tissue had nearly enveloped all exposed cartilage areas, obscuring the gross observation of vascular bridging, though vascular formation was present in all cases on tissue sections. Unlike skin grafts, the PAT grafts did not allow for a clear demarcation between viable and necrotic tissue on the dates of gross observation. Histological analysis also indicated areas of cellular attrition, yet a membranous structure persisted. Consequently, the precise determination of graft survival rates was challenging. Notably, granulation tissue formation was significantly accelerated by day 14, suggesting that even in the absence of vascular bridging, the grafts likely contributed to wound healing through their biomaterial properties, including stem content.

The aforementioned explanation has been incorporated into the manuscript in sections 3.3 “PAT graft survival,” 3.4.1 “Granulation tissue growth,” and 4.8.2 'Comparative Analysis with Other Treatment Modalities.”

4. What's the mechanism of action of PAT for granulation tissue formation, it's effect as a viable graft or it's effect when necrosis occurs and intra cellular mediators such as growth factors release? If first hypothesis is correct please mention what's the mechanism of PTA viability.

(Response)

We are grateful for your guidance. To address this, we have introduced a new subheading, 4.7 “Mechanisms underlying PAT graft viability.” This section is dedicated to elucidating the inferred biological mechanisms that contribute to the viability of PAT grafts.

5. You stated: "The open wound area, as determined from photographs, was significantly larger in the PAT 345 group than in the control group on day 7 (P < 0.05) (Fig 4A)". Does this mean wounds on both groups were not similar?.

(Response)

We apologize for the confusion and thank you for highlighting the ambiguity in this part. Wound healing principally involves two mechanisms: wound contraction and re-epithelialization. Given that wound contraction can lead to hypertrophic scarring, optimal treatment strategies aim to minimize this process, particularly in humans, without impeding overall wound closure. In our study, the PAT group exhibited a transient delay in wound closure due to reduced contraction by day 7, and yet demonstrated an expedited re-epithelialization by day 14. Consequently, the total time to wound closure in the PAT group was found to be comparable to that of the control group by the 14th day. This suggests that the observed process in the PAT group may represent an ideal paradigm for wound healing.

For easier understanding, we have added explanations in sections 3.4.3 “Wound contraction evaluation” and 4.6 “Wound contraction control with PAT graft.”

6. What's the mechanism of viability of PAT graft.

(Response)

We have added a section, 4.7, “Mechanisms underlying PAT graft viability,” accordingly. This new section provides a comprehensive summary of the currently understood biological processes contributing to the efficacy of PAT grafts.

7. You stated "The effectiveness of PAT grafts can be attributed to two primary factors, namely vascular bridging during graft survival and the provision of three essential elements that promote wound healing and angiogenesis: progenitor/stem cells, ECM molecules, and growth factors/cytokines.". My questions is why skin graft cannot do this? To best of our knowledge skin graft cannot cover or take on bare areas such as bone without periosteum but you concluded PAT can. Please clearly stat what's the difference.

(Response)

Thank you for highlighting the missing explanation. There has long been a theory that full-thickness skin grafts exhibit a “bridging” capability, enabling them to extend over ischemic tissue. However, as you correctly note, the current consensus now holds that such wounds are generally not amenable to grafting. After a detailed review of the literature, we have discussed the mechanism underlying this phenomenon, integrating our findings into section 4.2, “Vascular bridging.”

Furthermore, we have expanded on the distinctions and advantages of PAT grafting in comparison to traditional skin grafting. This comparative analysis, particularly focusing on the unique aspects of PAT grafting, is presented in section 4.8.1, “Distinctive aspects relative to skin grafting.”

8. I think the manuscript is too long and includes parts that are unnecessary.

(Response)

We've tried to cut down on unnecessary parts, but the manuscript has still become longer due to some added sentences. If you still think the manuscript includes parts that are unnecessary, could you please specify which parts you think are not needed?

---

## [Decision Letter · Decision Letter 1]

2 Jan 2024

PONE-D-23-30878R1Perifascial Areolar Tissue Graft Promotes Angiogenesis and Wound Healing in an Exposed Ischemic Component Rabbit ModelPLOS ONE

Dear Dr. Miyanaga,

Thank you for submitting your manuscript to PLOS ONE. After careful consideration, we feel that it has merit but does not fully meet PLOS ONE’s publication criteria as it currently stands. Therefore, we invite you to submit a revised version of the manuscript that addresses the points raised during the review process.

We look forward to receiving your revised manuscript.

Kind regards,

Alessandra Giuliani

Academic Editor

PLOS ONE

Journal Requirements:

Reviewers' comments:

Reviewer's Responses to Questions

**Comments to the Author**

1. If the authors have adequately addressed your comments raised in a previous round of review and you feel that this manuscript is now acceptable for publication, you may indicate that here to bypass the “Comments to the Author” section, enter your conflict of interest statement in the “Confidential to Editor” section, and submit your "Accept" recommendation.

Reviewer #1: All comments have been addressed

Reviewer #2: All comments have been addressed

2. Is the manuscript technically sound, and do the data support the conclusions?

Reviewer #1: Yes

Reviewer #2: Partly

3. Has the statistical analysis been performed appropriately and rigorously? 

Reviewer #1: Yes

Reviewer #2: I Don't Know

4. Have the authors made all data underlying the findings in their manuscript fully available?

Reviewer #1: Yes

Reviewer #2: Yes

5. Is the manuscript presented in an intelligible fashion and written in standard English?

Reviewer #1: Yes

Reviewer #2: No

6. Review Comments to the Author

Reviewer #1: (No Response)

Reviewer #2: Dear Author

Hello,

I read revised manuscript. The desired corrections have been made, but the writing style of the article is confusing for the reader. This is due to the lengthy and repetitive sentences, which make it tiresome to read and difficult to understand the main concept of the article.

7. PLOS authors have the option to publish the peer review history of their article (what does this mean?). If published, this will include your full peer review and any attached files.

Reviewer #1: **Yes: **Agarbati Silvia

Reviewer #2: **Yes: **Mehdi Ayaz

---

## [Author Response · Author response to Decision Letter 1]

30 Jan 2024

Response to the Reviewer’s comments 

Dear Editor;

Thank you for evaluating our manuscript and for your encouraging comments and appreciation of our work. Our manuscript has benefited from your comments and the reviewers’ guidance and thoughtful suggestions. We have read the comments provided by you and the reviewers and revised the manuscript accordingly. Please find our point-by-point responses to all the comments below. 

Reviewer #2: 

I read revised manuscript. The desired corrections have been made, but the writing style of the article is confusing for the reader. This is due to the lengthy and repetitive sentences, which make it tiresome to read and difficult to understand the main concept of the article. 

(Response)

Thank you for pointing out this issue. We agree that the sentences were long and difficult to understand because of repetitive text. We modified the text to avoid repetition and improved understandability. Consequently, the word count has been reduced from 6340 to 5407 words. Thank you again.

---

## [Editor Report · Decision Letter 2]

2 Feb 2024

Perifascial Areolar Tissue Graft Promotes Angiogenesis and Wound Healing in an Exposed Ischemic Component Rabbit Model

PONE-D-23-30878R2

Dear Dr. Miyanaga,

We’re pleased to inform you that your manuscript has been judged scientifically suitable for publication and will be formally accepted for publication once it meets all outstanding technical requirements.

Kind regards,

Alessandra Giuliani

Academic Editor

PLOS ONE
---

## [Editor Report · Acceptance letter]

8 Feb 2024

PONE-D-23-30878R2 

PLOS ONE

Dear Dr. Miyanaga, 

I'm pleased to inform you that your manuscript has been deemed suitable for publication in PLOS ONE. Congratulations! Your manuscript is now being handed over to our production team.

Kind regards, 

on behalf of

Dr. Alessandra Giuliani 

Academic Editor

PLOS ONE